# Latent Diffusion Models to Enhance the Performance of Visual Defect Segmentation Networks in Steel Surface Inspection

**DOI:** 10.3390/s24186016

**Published:** 2024-09-18

**Authors:** Jon Leiñena, Fátima A. Saiz, Iñigo Barandiaran

**Affiliations:** Vicomtech Foundation, Basque Research and Technology Alliance (BRTA), 20009 Donostia, Spain; fsaiz@vicomtech.org (F.A.S.); ibarandiaran@vicomtech.org (I.B.)

**Keywords:** defect segmentation, data augmentation, stable diffusion, industrial manufacturing, quality control

## Abstract

This paper explores the use of state-of-the-art latent diffusion models, specifically stable diffusion, to generate synthetic images for improving the robustness of visual defect segmentation in manufacturing components. Given the scarcity and imbalance of real-world defect data, synthetic data generation offers a promising solution for training deep learning models. We fine-tuned stable diffusion using the LoRA technique on the NEU-seg dataset and evaluated the impact of different ratios of synthetic to real images on the training set of DeepLabV3+ and FPN segmentation models. Our results demonstrated a significant improvement in mean Intersection over Union (mIoU) when the training dataset was augmented with synthetic images. This study highlights the potential of diffusion models for enhancing the quality and diversity of training data in industrial defect detection, leading to more accurate and reliable segmentation results. The proposed approach achieved improvements of 5.95% and 6.85% in mIoU of defect segmentation on each model over the original dataset.

## 1. Introduction

In recent times, the manufacturing industry has witnessed a significant shift towards automation, encompassing a wide range of processes from product assembly to resource planning, and evolving from basic repetitive tasks to intricate decision-making operations. Quality inspection, in particular, has seen a notable transition with the adoption of advanced algorithms and machine vision systems, which undertake duties previously reliant on human scrutiny, facilitated by the advent of visual computing technologies [1].

The shift from human quality inspection to computer-automated quality inspection has not only enhanced accuracy and efficiency, but has also allowed for continuous monitoring, ensuring that production standards are met at all times, from the initial stages of manufacturing to the final product rollout.

The presented work revolves around steel’s quality control: due to its usual manufacturing workflows, which commonly require mechanical and thermal handling, superficial defects may arise at multiple stages and pertain to different categories. As a result, quality assurance involves the analysis, detection, and rejection of the possible superficial, structural, and dimensional defects.

As aforementioned, quality control’s automation enables the inspection of virtually all manufactured elements with high precision, in order to live up to the quality standards of an increasingly demanding market. To achieve such levels of precision and inspection rate, deep learning-based models have become the go-to approach in automated quality control [2]. However, the development of robust, precise, and fast models requires large amounts of industrial, human annotated data, which is especially complicated and expensive, and sometimes even impossible. Furthermore, to maintain profitability, a company’s production largely needs to be free from defects, rendering such imperfections rare and leading to highly imbalanced datasets. Using such datasets can hinder model performance, causing a bias towards the more frequently occurring class.

These challenges motivate research into generative models as a viable solution to create high-quality, high-fidelity images that can mitigate data scarcity and improve model performance. Several techniques have been successfully employed to fill this gap, such as conventional data augmentation, which applies transformation operations to the images such as rotation, scaling, or flipping. However, this method has a limitation in that it does not provide new features to the generated images. Traditional data augmentation also has its problems relating to the trial-and-error scenario it presents until the required configuration of transformations that the model really needs to improve its performance is found, which means an increase in development time [3].

As a means to resolve this issue, generative models that can create high-quality and high-fidelity images pose as an interesting alternative, particularly in light of the latest advances in the industrial field, showcased by research utilizing open models like stable diffusion [4,5].

### 1.1. Defect Detection Using Deep Learning

The development of deep learning has revolutionized the way industrial challenges are approached thanks to its ability to analyze complex data with remarkable accuracy and efficiency [6]. Among the many use cases of deep learning algorithms for superficial inspection, the most common ones are classification [7], region-based defect detection [8], and semantic segmentation [9,10]. This last application has become a popular research topic, given its capabilities to show clear, pixel-level object categories in images.

Recent studies have focused on the use of deep segmentation networks in production systems in order to achieve the highest possible precision regarding location and shape of defects [11]. Authors performing research in the area have proven the capabilities of deep learning models to perform computationally efficient, fast, and precise segmentation of images, in the context of industrial quality control [12]. For instance, models using architectures such as DeepLabV3+ [13] or FPN [14] have shown astonishing capabilities in the task of defect segmentation of metallic surfaces, reaching mean Intersection over Union (mIoU) metrics of 89.52% and 90.68%, at 35.14 and 30.77 frames per second (FPS), respectively [15]. The mIoU metric measures the accuracy of the segmentation by calculating the average IoU across all classes, where IoU is the ratio of the overlap between the predicted segmentation and the ground truth to their union. IoU values range from 0 to 1, where 0 means no overlap, and 1 means perfect overlap. In the segmentation task, this calculation is performed on a pixel-by-pixel level [16]. FPS indicates the processing speed of the model, representing the number of frames processed per second.

Other authors [17] have shown how the same DeepLabV3++ architecture, using the well-known ResNet50 backbone, can obtain comparable results to the previous study concerning Intersection over Union and precision metrics. Comparative analyses, including those by Sime et al. [15] and Liu et al. [17], highlight how the DeepLabV3++ and FPN architectures perform better than popular architectures such as U-Net, SegNet, and FCN, among others.

### 1.2. Advances in Industrial Data Synthesis

A significant challenge lies in the acquisition of high-quality images, which is often hindered by the intricate shapes of components, materials with high reflectivity or glossiness, and the changing lighting conditions typical of an industrial setting. Consequently, the development of a reliable and robust machine vision system is invariably a complex undertaking.

Moreover, procuring defective product samples is not always feasible. Given the superior production quality maintained by manufacturers, the majority of the components are devoid of defects. It is also important to note that the rapid pace of production can make it challenging to identify faulty parts. This issue is frequently encountered when gathering datasets for training models designed to detect defects in industrial components. As a result, the creation of annotated synthetic datasets is emerging as a promising field of study within the machine vision community.

There are multiple techniques that can be used to generate synthetic data, ranging from traditional methods that use image processing to generate new samples, to deep learning-based techniques that use generative adversarial networks, to state-of-the-art diffusion models, among others [18]. The potential of synthetic data in improving the robustness of models used in defect detection tasks [19], in addition to the associated reduction in cost of data obtention, make it a highly interesting topic for industrial companies.

Manufacturing defects often times follow a “long-tail” distribution [18], meaning that there are large amounts of instances for certain classes of defects, while others seldom happen. This renders the models’ generalization capabilities less effective in the detection task. As a result, limiting data augmentation to traditional augmentation methods, which include rotations, flips, intensity transformations, etc., is not the most suitable solution; although traditional methods do aid in improving model generalization capabilities, they can only expand on existing data, making it more complicated to address the augmentation of less-common faulty patterns. To overcome such needs of generating new data points which will address the least common patterns in the existing datasets, most efforts were focused on studying the use of GANs for the task of industrial data synthesis. This continued until the appearance of diffusion models, which have proven to be capable of generating higher quality and more realistic images [20]. These generative models have the ability to generate completely new synthetic data with a degree of randomness, hence creating rich and diverse augmentations. Specifically, diffusion models sample a probability distribution of data points through an iterative process, which begins with randomly-generated grey noise, generating completely new, feature-rich images each time. It is this feature that makes them especially interesting for the aforementioned task of data augmentation of imbalanced datasets, even more so when compared to traditional data augmentation processes.

According to the most recent findings, diffusion models, specifically latent diffusion models (LDM) like stable diffusion [21], have proven to be vastly superior to GANs in the task of image generation of industrial surface defects [18]. Moreover, authors such as Hu et al. and Valvano et al. have advanced the generative capabilities of LDMs by applying various conditioning techniques to enhance adaptation to complex domains, such as the industrial domain [4,5]. By tuning the generative models with added conditioning via anomaly masks or hypernetworks, these researchers are able to obtain highly realistic, varied, and large synthetic datasets, which can be used to train new models for the adjacent proposed tasks of anomaly detection and semantic segmentation.

### 1.3. Contributions

This paper analyzes the use of LDMs, specifically stable diffusion, for data synthesis in the creation of high-quality synthetic samples in order to balance and enrich industrial surface defect datasets. Our main contribution lies in implementing and validating a new trained model based on the LoRA [22] technique for generating new images for the dataset, and using them to improve the performance of semantic segmentation networks. We performed a comparison of the benefits of adding synthetic images to the training set versus those obtained when a network is trained with traditionally augmented data. We validated the approach with well-known architectures such as DeepLabV3++ and FPN, using the public NEU-seg [23] dataset. The results confirm that the use of data augmentation by synthetic generation employing stable diffusion not only improves segmentation accuracy, but also demonstrates the repeatability of the method on different network architectures. The synthetic generated dataset is publicly published in order to allow the replication of the obtained results, and to encourage its use by the research community.

## 2. Materials and Methods

### 2.1. NEU-Seg Dataset

The dataset used to carry out the experiments was the widely known NEU-seg dataset [23]. This dataset contains 3630 grayscale images of 200 × 200 pixels for training, with pixel-level annotations of three different defect classes: inclusion, patches, and scratches, which make up 33.7% (1460), 33.4% (1448), and 32.9% (1429) of the training set, respectively [15]. Figure 1 depicts the raw images of the dataset and masks for each defect type. The top row displays the raw grayscale images of these defects, while the bottom row presents the corresponding ground truth segmentation masks, where patches are marked in blue, scratches in red, and inclusions in green. Scratches are linear defects caused by physical abrasion on the metal surface, clearly visible as red lines in the segmentation masks. Inclusions are non-metallic particles trapped within the metal during the manufacturing process, appearing in green in the masks. Patches refer to areas where the surface coating or material is missing or altered, shown in blue in the segmentation masks. This comprehensive labelling allows for precise identification and study of each defect type within the dataset, facilitating the development and evaluation of defect detection algorithms. The used distribution for training, validation, and test sets is presented in the Section 3.

### 2.2. Experimental Design

Our approach leveraged the state-of-the-art stable diffusion model to generate synthetic images of metal surfaces to augment the public NEU-seg dataset, addressing the challenges associated with data scarcity and class imbalance in defect segmentation tasks. Our method follows the process shown in Figure 2, where the inputs and outputs of each step are mentioned, as well as their interconnection with other experimental steps. All the steps are described in more detail below:Step 1: We begin by fine-tuning the pre-trained state-of-the-art stable diffusion model using the LoRA technique [22] on the three different classes of the NEU-seg dataset. The fine-tuning process is performed on the three defect classes of the NEU-seg dataset: scratches, cracks, and pitting. Through training the model on these specific defect categories, we aim to generate synthetic images that are visually not only realistic, but are also representative of the variations in defect characteristics within each class.Step 2: Once the synthetic images are generated, we need to obtain accurate defect masks to use them as new data for training segmentation models. For this task, we employ a DeepLabV3+ segmentation network that has been trained on the original NEU-seg dataset. The DeepLabV3+ network is selected due to its high performance in pixel-wise segmentation tasks, in particular for complex surface textures typical of metallic materials. The result of this step is defect masks for each synthetic image, which are manually reviewed to ensure that the necessary ground truth is obtained to retrain the segmentation models. This step allows the optimization of the annotation process by using the model output masks and refining them when necessary, rather than annotating all images from scratch.Step 3: With the synthetic images and their corresponding defect masks, we augmented the NEU-seg dataset by combining these synthetic samples with the original dataset. We experiment with different ratios of synthetic images to the original ones (e.g., 20–80%, 50–50%) to understand how different levels of dataset enrichment affect segmentation performance.Step 4: With the augmented and enriched dataset, we proceed to retrain the DeepLabV3+ segmentation network with the aim of improving the network’s generalization ability and understanding how dataset enrichment affects the segmentation results.Step 5: To gauge the broad utility of our method, we extend the evaluation to an alternative segmentation architecture, the FPN network, chosen for its strong performance shown in the task of metallic surface segmentation [15]. The FPN is trained under the same conditions as the DeepLabV3+, using the same augmented NEU-seg dataset.

**Figure 2 sensors-24-06016-f002:**
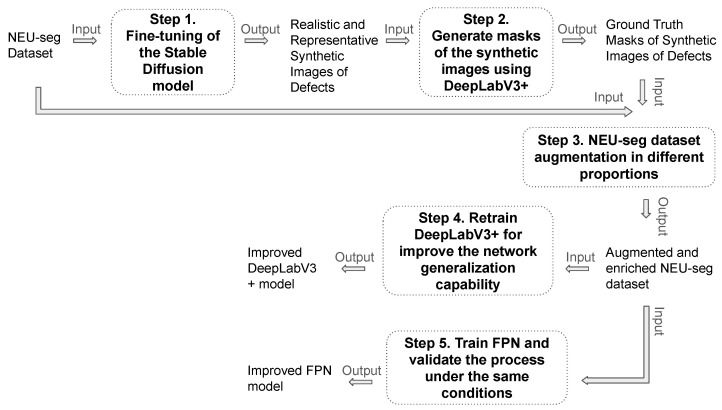
Experimental process followed, detailing the inputs and outputs of each step, and their interconnection with the other stages of the experiment.

### 2.3. Conventional Data Augmentation

The training set of the NEU-seg dataset used in the experiments underwent a conventional data augmentation approach, which consists of randomly applying a set of geometrical and photometric transformations to the original images. These increase the diversity of the dataset, which helps models to generalize better and become more robust [24]. The augmentations applied have been the following, each with the probability assigned to it. These probabilities determine the likelihood that a specific augmentation will be applied to an image, adding randomness and variability to the dataset, seeking to enhance model’s ability to generalize:Horizontal/vertical Flip transformation: simulates various defect orientations with a probability of 0.5.Addition of Gaussian noise: introduces texture variations with a probability of 0.2, in search of enhanced noise robustness.Contrast-limited adaptive histogram equalization: improves contrast for better detection of fine details, applied with a probability of 1.Brightness modification: simulates diverse lighting conditions, applied with a probability of 1.Gamma modification: adjusts tone intensity, applied with a probability of 1.Sharpening: enhances defect edges and details, applied with a probability of 1.Blurring: varies image sharpness, mimicking different focus levels, to improve model robustness across varying focus conditions, applied with a probability of 1.

### 2.4. Image Synthesis with Stable Diffusion

The stable diffusion architecture consists of three main parts: a variational autoencoder (VAE), a U-Net network, and an optional text encoder [21]. The VAE encoder compresses the image from pixel space to a lower dimensional latent space, capturing a more fundamental semantic meaning of the image. Then, Gaussian noise is iteratively applied to the compressed latent representation during forward diffusion, after which the U-Net block handles the backward diffusion process to obtain a realistic output.

Finally, the VAE decoder generates the final image by converting the representation back to pixel space. The denoising step can be flexibly conditioned to a text string, an image, or another modality. To condition the text, the pre-trained CLIP ViT-L/14 [25] text encoder is used to transform the text prompts to an embedding space. Figure 3 depicts the aforementioned architecture.

### 2.5. Fine-Tuning Stable Diffusion with LoRA

LoRA (low-rank adaptation) appeared first as a fine-tuning technique for large language models [22], which greatly reduced the trainable parameters of said models by adding trainable layers to the Transformer attention blocks. As shown in Figure 3, stable diffusion has cross-attention layers that are in charge of relating the image representations with the text prompts used to describe them. Hence, in the case of stable diffusion, LoRA fine-tuning works by injecting trainable matrices in these cross-attention blocks, enabling targeted adjustments with minimal computational overhead and without altering the underlying model architecture.

One of the key advantages of LoRA is that it merely fine-tunes the model, leaving the original architecture untouched. This ensures that the model retains its pre-trained capabilities while allowing for efficient adaptation to new tasks or domains. The non-intrusive nature of LoRA makes it a highly flexible and efficient method for fine-tuning, as it preserves the integrity of the original model while significantly reducing the number of parameters that need to be trained.

This fine-tuning technique vastly reduces the total amount of trainable parameters to adapt, GPU memory requirements, and the model’s weight and size. These advantages make LoRA an especially efficient and scalable approach for adapting models to new domains. In our experiments, LoRA was selected for fine-tuning due to its demonstrated effectiveness, as highlighted in the study by Zhong et al. [18], which compared various traditional and deep learning-based approaches to industrial data generation. The authors concluded that stable diffusion fine-tuned with LoRA outperformed other methods in terms of both image quality and generation efficiency, according to key metrics.

To explore the specific contributions of the LoRA technique within our experimental framework, we assessed its performance in generating images across different defect classes: inclusion, patches, and scratches. Table 1 provides the hyperparameters used for the fine-tuning process. The prompt used for image generation substituted <C> with the respective defect class. The results, as demonstrated in the subsequent analysis, underscore the efficiency and adaptability of the LoRA fine-tuning approach, particularly in scenarios demanding high-quality image generation with constrained computational resources.

### 2.6. Defect Segmentation Using DeepLabV3+

DeepLabV3+ is a state-of-the-art semantic segmentation model, an extension of the DeepLabV3 model, which combines the strengths of both encoder-decoder structures and dilated convolutions [13]. The architecture was designed to extract features using atrous spatial pyramid pooling (ASPP) and then refine the segmentation results with a decoder module. The ASPP module captures multi-scale contextual information without losing resolution, which is crucial for semantic segmentation tasks. The decoder then combines the encoded feature maps with higher-resolution, lower-level feature maps, until it reaches the same dimensions as the input image. Figure 4 illustrates the DeepLabV3+ architecture [13].

The model’s output is an image, corresponding to the pixelwise classification of the image into a particular class. As described in Section 2.1, the used dataset contains three defect classes, plus the “background” or “non-defect”. The evaluation metric of Intersection over Union was chosen to better represent the results, since it is a measure of the overlap between predicted and ground-truth defect masks.

### 2.7. FPN for Multi-Scale Defect Segmentation

The FPN [14] architecture leverages a bottom-up pathway and a top-down pathway within a convolutional neural network. In the bottom-up pathway, features are extracted at multiple scales through a series of convolutional and pooling operations. The top-down pathway then upsamples these features to produce high-resolution feature maps, which are combined with corresponding features from the bottom-up pathway through lateral connections. This allows for the integration of both semantically rich features from deeper layers and fine-grained spatial information from shallower layers, leading to improved segmentation accuracy, especially for small objects.

As opposed to the previously trained DeepLabV3+ architecture based models, which use Atrous Spatial Pyramid Pooling (ASPP) to capture multi-scale context, FPN constructs a pyramid of feature maps at different scales by upsampling deeper features and combining these with the corresponding features of earlier layers through lateral connections. In the refinement stage, DeepLabV3+ makes use of the atrous convolutions to further refine the segmentation results; whereas FPN utilizes simple convolutional layers to incorporate the multi-scale features to generate the segmentation mask.

## 3. Results

This section outlines experiments evaluating the use of state-of-the-art generative models, such as stable diffusion, to enhance datasets for training steel defect segmentation models. We first document the variation in segmentation performance for various synthetic-to-real image ratios in the dataset. Secondly, we validate whether the obtained metrics with the optimal image ratio using the DeepLabV3+ architecture are replicable using another architecture such as FPN.

### 3.1. DeepLabV3+ Results on Original NEU-Seg Dataset

In this first experiment, we trained the DeepLabV3+ defect segmentation model using only the original images from the NEU-seg dataset, using training, validation, and test splits following the image distribution show in Table 2. For validation purposes, we utilized 30% (equivalent to 1089 samples) of the training set. The dataset also included 840 images, each with per-pixel annotations, intended for testing. All experimental outcomes were documented based on the test set.

The hyperparameters chosen for training the model were set according to the ones reported by Sime et al. [15].

Figure 5 shows some visual detection results obtained with this model and the NEU-seg original dataset. In this first experiment, we obtained a mean IoU of 71.31%, which will be used as the baseline benchmark for the forecoming experiments.

### 3.2. Stable Diffusion Optimized with LoRA for Synthetic Image Generation

At this stage, to increase the dataset significantly and to achieve an increase in mIoU, we used stable diffusion to generate training samples. For the LoRA training process of the generative model, the original training set of NEU-seg was used, although separated by defect classes. This way, the stable diffusion model was used to generate as many samples of each class as required in order to keep the dataset balanced.

The synthesized images correctly capture the characteristics and shapes of each defect class, conveniently adding variability in shape, size, contrast, etc. Nonetheless, the generated samples require annotation in order to be used as training samples in the dataset for the segmentation network. To achieve this, the generated images were annotated by running inference on the DeepLabV3+ model trained in the aforementioned experiment, and carefully revised visually one by one to ensure the quality of the self-annotated masks, which were corrected in case of inaccuracies, ensuring the quality of the newly generated masks and their validity for further use as ground truth. Some results of this process can be seen in the set of images shown in Figure 6.

### 3.3. Effect of Synthetic Image Ratio in Training Dataset

This experiment tests out the effect of using different ratios of synthetic and original images in the training dataset of the DeepLabV3+ model. Ten different instances of the model were trained for semantic segmentation, where each instance was trained on a different dataset than the rest, varying both the number of original images and synthetic images in the dataset. Consequently, the ratio of synthetic versus real images on the training set changes, but the validation and test datasets have been left untouched. Table 3 shows the different amounts and ratios of original and synthetic images in each training dataset, as well as the mIoU reached by the segmentation model on the test dataset.

The results clearly show an improvement in model performance when the training dataset is augmented with images generated by stable diffusion. Additionally, it is also remarkable that it does so until the proportion of original images in the dataset falls below 50% of the total image count. Table 3 shows that the optimal performance is achieved when synthetic images constitute 37% of the training dataset.

This behaviour occurs because the synthetic images help stabilize the dataset by introducing more examples of features that were underrepresented in the original dataset. We believe that adding synthetic images is beneficial up to a certain point, as they enhance the model’s generalization capabilities. However, there is a threshold; if exceeded, the generated images might introduce erroneous or confusing features. This could, instead of aiding the model, hinder its generalization by distorting the original dataset’s distribution. Defining a specific optimal threshold for the proportion of synthetic images is challenging and cannot be generalized to other applications or domains, just as it is difficult to determine an exact number of images needed to train any model. An iterative approach is necessary, gradually increasing the dataset until a convergence point is reached.

The performance gain of the model can be perceived in Figure 7, which showcases the comparison between the original image’s ground truth annotations, the DeepLabV3+ model’s predictions when trained on the unaugmented NEU-seg dataset, and the predictions of the DeepLabV3+ model when the training dataset of the model is augmented with 37% stable diffusion generated images. By comparing the different columns of Figure 7, it can be seen how the prediction improves before and after the training dataset is augmented with the proposed method.

### 3.4. Validation of the Experiments Using FPN Architecture

In order to replicate the results obtained in the previous experiments, to reaffirm and validate the benefits of data augmentation using stable diffusion, new semantic segmentation models were trained using the FPN architecture. The dataset augmentation was performed with different proportions of synthetic data and original images in the training set, as performed with DeepLabV3+. Table 4 shows the obtained metrics for each image percentage combination.

After conducting said experiments, the mean Intersection over Union (mIoU) achieved with the using original dataset with traditional data augmentation methods for training was 72.39%, whereas training the model with the optimized synthetic dataset resulted in an improved mIoU of 76.7%. Figure 8 shows the mIoU values of both models, FPN and DeeplabV3+, comparing the results obtained in the test set with each model trained with the different datasets mentioned above, where the differences in the values mentioned can be appreciated in a more visual way. The consistent improvement in mean Intersection over Union (mIoU) observed across multiple segmentation networks, which proves the benefits of augmenting datasets with synthetic images.

## 4. Discussion

This work demonstrates that using stable diffusion to generate synthetic images significantly improves the performance of defect segmentation models for industrial inspection tasks. Specifically, we observed enhancements in the mean Intersection over Union (mIoU) for both DeepLabV3+ and FPN models when optimal ratios of synthetic to real images were used. These improvements are indicative of an enhancement in the models’ capacity to detect and segment defects on metallic surfaces, as a result of the incorporation of high-quality synthetic data into their training sets.

Our findings align with previous research in the area that has proven the potential of synthetic data to improve the performance of defect segmentation models [5,19]. Past studies [19] show the effectiveness of using synthetic data generated by GANs for similar tasks. However, this work differs in several key aspects. First, we have used stable diffusion, fine-tuned with LoRA, for the image generation task, showcasing the versatility of this foundation model. Finally, we have also provided the impact of different ratios of synthetic to real images on model performance, providing valuable insights into the optimization of the use of synthetic data in practical applications.

The observed improvements in model performance can be attributed to multiple factors. The diversity of the images generated by stable diffusion likely contributes to better feature learning and generalization, enabling the models to effectively capture the variability of defects. In addition to that, the high quality of the synthetic images likely reduced the domain gap between synthetic and real images, which, added to the selection of the optimal ratio of synthetic to real images, may have provided a balance between diversity and realism, avoiding overfitting and enhancing the models’ detection and generalization capabilities.

## 5. Conclusions

As a conclusion, we can assert the effectiveness of using synthetic images generated by stable diffusion to improve the performance of semantic segmentation models for industrial defect detection. By fine-tuning stable diffusion with the LoRA technique and using the resulting model to augment the NEU-seg dataset, we achieved a notable improvement in the mean Intersection over Union (mIoU) for both DeepLabV3+ and FPN architectures. Specifically, the mIoU for DeepLabV3+ increased from a baseline of 71.3% to 76.2% when trained on a dataset with 37% synthetic images, while the mIoU for FPN increased from 72.3% to 76.7%. These results underscore the potential of state-of-the-art diffusion models to address the data scarcity and imbalance challenges in industrial defect detection tasks.

Our work contributes to the growing body of research on synthetic data generation for industrial applications.The insights from our experiments on the impact of different synthetic-to-real image ratios provide valuable guidelines for practitioners in optimizing the use of synthetic data for their specific applications. This can lead to more efficient, cost-effective, and reliable automated defect inspection systems in the manufacturing industry.

It would be interesting in future studies to test out other architectures, such as Transformer-based architectures, through similar experimentation, given their state-of-the-art results and promising segmentation capabilities [15]. It would also be useful to compare the proposed data augmentation method with other relevant state-of-art methods on NEU-seg using the same test set. Additionally, to further explore the performance of stable diffusion and the impact of generating synthetic images, other datasets should also be explored. This could provide additional insights into the versatility and applicability of stable diffusion in different contexts and tasks, potentially leading to even more significant improvements in model performance and robustness.

## Figures and Tables

**Figure 1 sensors-24-06016-f001:**
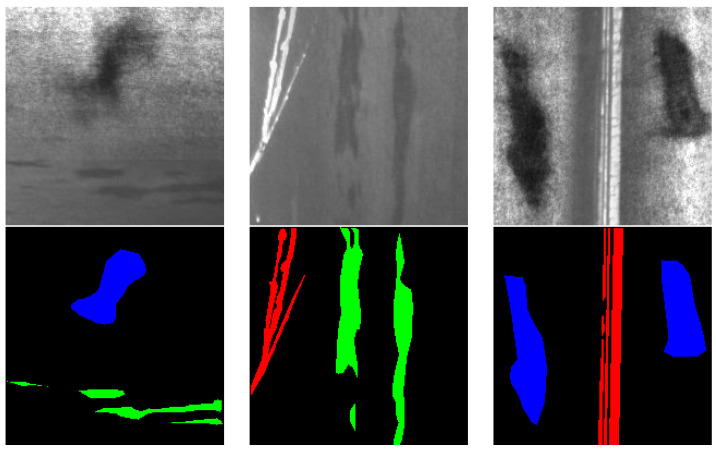
Raw images and ground truth masks for NEU-seg dataset [23], where patches are marked in blue, scratches in red and inclusions in green.

**Figure 3 sensors-24-06016-f003:**
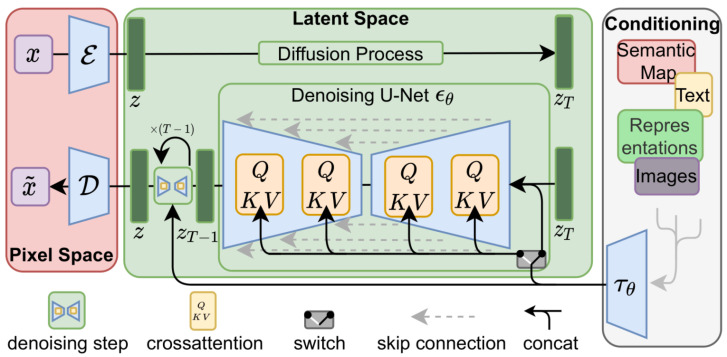
Architecture diagram of the stable diffusion network [13].

**Figure 4 sensors-24-06016-f004:**
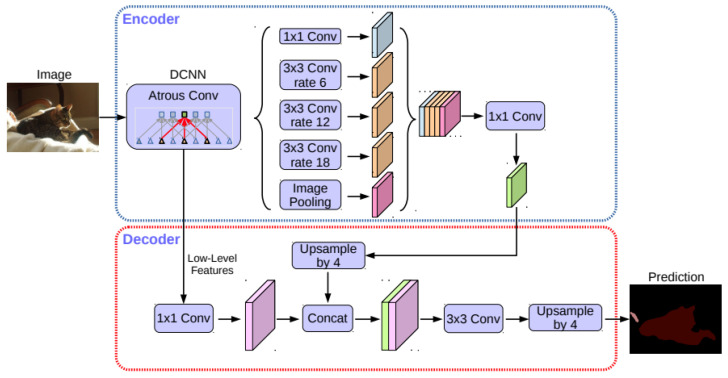
Architecture diagram of the DeepLabV3+ network [13].

**Figure 5 sensors-24-06016-f005:**
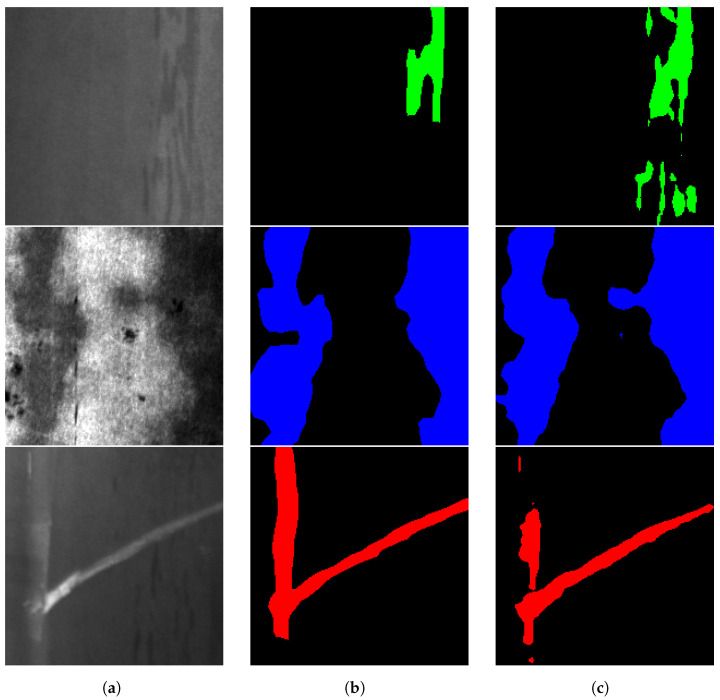
Visual results of the DeepLabV3+ network trained on the original data set, where the three types of defects mentioned above are shown under the same color scheme. (**a**) Raw surface images; (**b**) ground truth defect masks; and (**c**) predicted masks.

**Figure 6 sensors-24-06016-f006:**
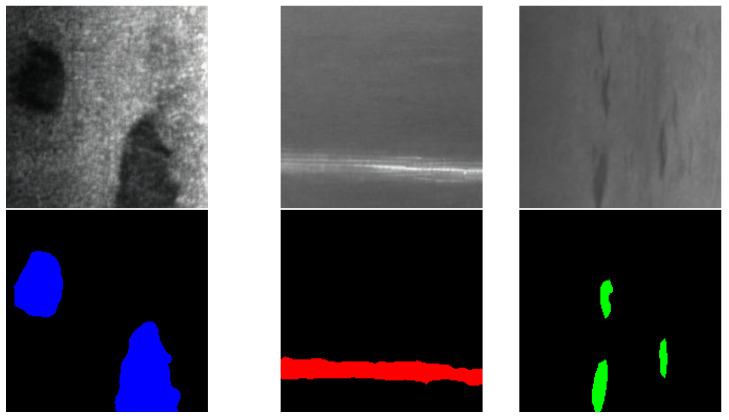
Synthetic raw surface images and DeepLabV3+ ground truth defect masks, where patches are marked in blue, scratches in red and inclusions in green.

**Figure 7 sensors-24-06016-f007:**
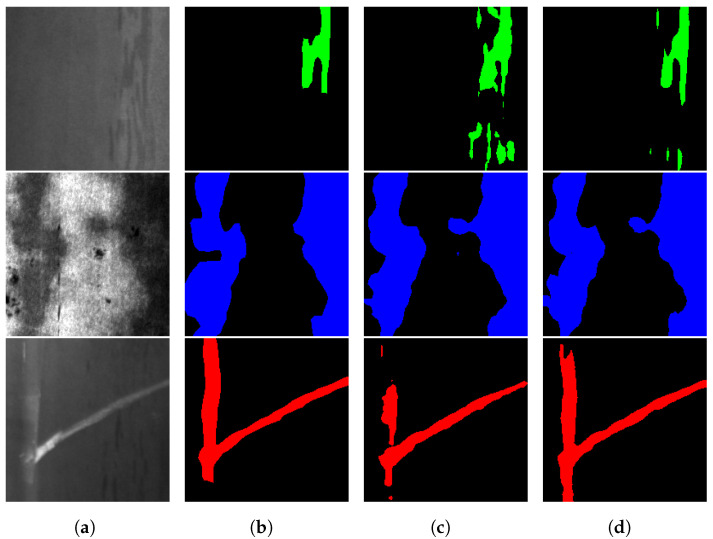
Segmentation results of two instances of the DeepLabv3+ network, where patches are marked in blue, scratches in red and inclusions in green. (**a**) Raw images; (**b**) ground truth masks; (**c**) predictions with the original NEU-seg dataset; and (**d**) predictions adding synthetic images.

**Figure 8 sensors-24-06016-f008:**
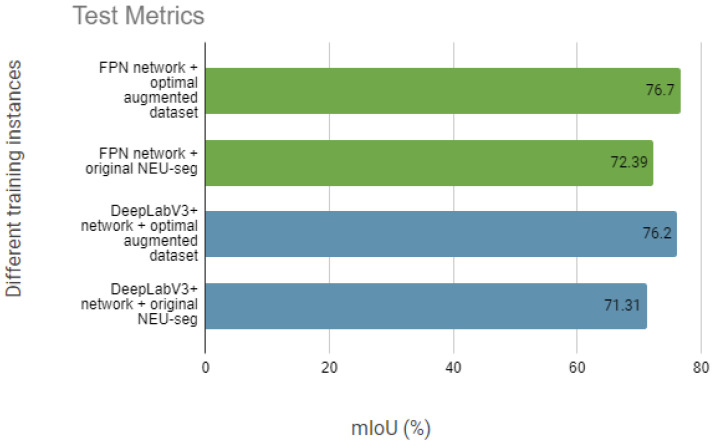
Mean of IoU (%) during evaluation with the original NEU-seg dataset and the original set plus synthetic images generated by stable diffusion with a 37% ratio.

**Table 1 sensors-24-06016-t001:** Hyperparameters used for fine-tuning stable diffusion with the LoRA technique.

Hyperparameters Used in the LoRA Fine-Tuning	
Parameter	Value
Precision	Mixed-precision FP-16
Batch size	4
Steps	5000
Initial learning rate	1 × 10^−4^
Learning rate scheduler	Constant
Learning rate warmup steps	0
Prompt	“a photo of superficial defect <C>”

**Table 2 sensors-24-06016-t002:** Dataset splits for semantic segmentation task using NEU-seg.

Dataset Split of NEU-Seg	
Dataset Split	Number of Images
Train	2541
Validation	1089
Test	840

**Table 3 sensors-24-06016-t003:** Different ratios of synthetic images in the composition of the training dataset and respective mean IoU values (%) on the test dataset.

Distinct Training Instances of DeepLabV3+ Models				
Number of Training Set Original Images	Number of Training Set Synthetic Images	Total Number of Training Set Images	Synthetic Image Ratio (%)	Mean IoU on Test Set (%)
2541	0	2541	0	71.31
2541	500	3041	16	74.94
2541	1000	3541	28	75.82
**2541**	**1500**	**4041**	**37**	**76.2**
2541	2541	5082	50	75.60
2000	2541	4541	56	75.37
1500	2541	4041	63	68.29
1000	2541	3541	72	68.07
500	2541	3041	84	66.12
0	2541	2541	100	63.8

**Table 4 sensors-24-06016-t004:** Different ratios of synthetic images in the composition of the training dataset and respective mean IoU values (%) on the test dataset.

Distinct Training Instances of FPN Models				
Number of Training Set Original Images	Number of Training Set Synthetic Images	Total Number of Training Set Images	Synthetic Image Ratio (%)	Mean IoU on Test Set (%)
2541	0	2541	0	72.39
2541	500	3041	16	73.11
2541	1000	3541	28	76.55
**2541**	**1500**	**4041**	**37**	**76.7**
2541	2541	5082	50	76.57
2000	2541	4541	56	55.16
1500	2541	4041	63	47.87
1000	2541	3541	72	56.57
500	2541	3041	84	52.84
0	2541	2541	100	66.9

## Data Availability

The raw data supporting the conclusions of this article will be made available by the authors on request.

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
