# Peer review of "Latent Diffusion Models to Enhance the Performance of Visual Defect Segmentation Networks in Steel Surface Inspection"

_sensors, 2024, doi:10.3390/s24186016_

Round 1

Reviewer 1 Report

Comments and Suggestions for Authors

This manuscript presents the use of LoRA technology to fine-tune stable diffusion and evaluates the impact of varying proportions of synthetic-to-real image training sets. The proposed method achieves improved results. However, there appear to be several issues with the experimental results and writing. Below are the detailed comments:

1The clarity of the images is insufficient, and there is a lack of detailed analysis and explanation of the image content. It is recommended to improve image quality and provide a thorough discussion of the content to enhance the overall presentation.

2In Tables 3 and 4, the Synthetic Image Ratio should be 100, not 1000, when original images are not used. This correction is necessary for accurate representation of the data.

3The overall structure of the paper requires optimization. Specifically, the dataset introduction should be moved to the experimental section, while the introduction of FPN should be placed in the methods section to enhance the paper's logical flow and readability.

4The introduction to related work is insufficiently comprehensive and does not clearly establish the research motivation of this paper. It is recommended to expand the discussion of relevant literature and explicitly articulate its logical connection to the work presented in this paper.

5Including literature citations in the figure and table titles is not ideal. It is advisable to move the citations to the figure or table captions to maintain clarity and conciseness in the titles.

6The description of the core method in the paper is insufficiently detailed and lacks logical coherence, failing to adequately highlight the method's advantages. It is recommended to provide a more comprehensive and detailed explanation of the method.

7In Section 3.1, experimental data are presented directly without a clear description of the experimental procedure. It is recommended to provide a detailed account of the experimental design and implementation steps in this section, so that readers can fully understand the data acquisition and processing methods.

8Section 3.2 does not adequately highlight the method proposed in this paper; the current title 'Stable Diffusion' does not reflect the LoRA method presented. It is advisable to revise the title to clearly convey the uniqueness and innovation of the proposed method.

9The analysis in the experimental section is insufficient and fails to effectively demonstrate the superiority of the proposed method. It is recommended to enhance the in-depth analysis of the experimental results and clearly compare the advantages of this method relative to others.

10The evaluation metric mIoU used in the experiments is not introduced, making it difficult for readers to understand its importance in the experimental context. It is recommended to include a description of this metric to clarify its relevance and significance.

11The experimental section does not compare the proposed method with current SOTA techniques, which limits the assessment of its relative performance. It is advisable to include comparisons with existing SOTA methods to comprehensively demonstrate the advantages of the proposed approach.

12The experimental section does not highlight the advantages of the LoRA technique. If experiments are conducted using images generated solely by stable diffusion, it is recommended to explore the results in this context to assess the specific contributions of the LoRA technique.

Comments on the Quality of English Language

The English quality of this manuscript is generally good, and the exchange of ideas is clear and coherent.

Reviewer 2 Report

Comments and Suggestions for Authors

Amid the sustained advancements in computer vision across various industries, the task of data collection within professional domains has assumed paramount significance. This paper employs a diffusion model to generate synthetic images and bolster the robustness of visual defect segmentation, constituting a highly significant and meaningful work.

Regarding this paper, there are some areas that require further explanation from the authors:

1. The authors mentioned in the article that there are significant differences between traditional data augmentation methods and diffusion model-based data synthesis methods. However, the authors did not explain the role of traditional data augmentation methods in diffusion models or the comparison between traditional data augmentation methods and diffusion models. Please clarify this issue in the article.

2. Figure 5 presents both the synthesized image and the corresponding ground truth. In this paper, the author elaborates on the utilization of self-annotation for image annotation. Consequently, please provide a detailed account of the process through which the synthesized image undergoes self-annotation.

3. Upon analyzing the final model performance, the author observed that the synthesized image exhibited optimal efficacy at a specific scale. Consequently, we inquire whether there exists any underlying rationale for this phenomenon, and furthermore, what is the appropriate range within which the synthesized image should be confined when applying this methodology to alternative domains?

4. This paper initially utilizes the DeepLabV3+ segmentation network, trained on the original dataset, to obtain a defect mask. However, it is acknowledged that the DeepLabV3+ segmentation network may not exhibit high accuracy in the specific application domain of interest. Consequently, if the images segmented by this network are inaccurate, a concern arises regarding whether the enhanced dataset, which incorporates these segmented images, will contain an excessive number of negative samples.

5. Figure 7 shows the comparison of different deep learning models on different datasets, but there is no description of Figure 7 in the paper. Please add this part to the content.

Comments on the Quality of English Language

Language optimization can be carried out in some parts.

Round 2

Reviewer 1 Report

Comments and Suggestions for Authors

I am fine with the revision, but the clarity of Figures 4 and 8 still needs improvement.

Reviewer 2 Report

Comments and Suggestions for Authors

All my problems have been resolved.

Comments on the Quality of English Language

The overall English proficiency is good.